# The Effect of Asymmetrical Occlusion on Surface Electromyographic Activity in Subjects with a Chewing Side Preference: A Preliminary Study

**DOI:** 10.3390/healthcare11121718

**Published:** 2023-06-12

**Authors:** Yubing Zhang, Kun Liu, Zhengwei Shao, Chengqi Lyu, Derong Zou

**Affiliations:** 1Department of Stomatology, Shanghai Sixth People’s Hospital Affiliated to Shanghai Jiao Tong University School of Medicine, Shanghai 200025, China; 2Department of Rehabilitation Medicine, Shanghai Sixth People’s Hospital Affiliated to Shanghai Jiao Tong University School of Medicine, Shanghai 200025, China

**Keywords:** asymmetrical occlusion, sEMG, chewing side preference, masseter muscle, sternocleidomastoid muscle, gastrocnemius muscle

## Abstract

The relationship between asymmetrical occlusion and surface electromyographic activity (sEMG) in people with different chewing preferences is not clear. In this study, the 5 s sEMG changes in the masseter muscle (MM), sternocleidomastoid (SCM), lateral (LGA), and medial (MGA) gastrocnemius muscles were recorded in controls, and subjects with chewing side preference (CSP) during clench with bilateral (BCR), left (LCR), and right (RCR) posterior teeth placement of cotton rolls. The images of the middle 3 s were selected and expressed as the root mean square (unit: μV/s). The EMG waves of bilateral muscles were compared by computing the percentage overlapping coefficient (POC). Only the POCMM of the CSP showed gender differences at BCR and RCR. Between the control group and the CSP group, there were significant differences in the POCMM and the POCLGA at BCR. In addition, there was a significant difference in POCMM and POCSCM between the two populations in different occlusal positions. The change in the POCSCM correlated with the change in the POCMM (r = 0.415, *p* = 0.018). The experiment-induced asymmetrical occlusion showed that the altered symmetry of the MM correlated with the altered symmetry of the SCM. Long-term asymmetrical occlusion (i.e., CSP) not only affects MM but also has potential effects on other superficial muscles (e.g., LGA).

## 1. Introduction

The effect of different dental occlusion positions on body postural balance has been studied quite extensively in recent years [1,2,3,4,5]. Postural control is regulated by the central and peripheral nervous system, the visual vestibular system, and the musculoskeletal system. It has been found that the functional state of the stomatognathic system contributes to postural balance [6]. A reasonable account for the effect of occlusion on postural maintenance is the functional coupling between the masseter muscle and other postural muscles [4].

Studies have demonstrated that changes in occlusal positions can not only directly affect the surface electromyography (sEMG) of the masseter muscle [7], but also have different effects on muscles in other parts of the body. Recent studies have found that the functionally more symmetrical maxillary position resulted in a more symmetrical sternocleidomastoid contraction and less body sway [8,9]. Asymmetrical occlusal interference can produce changes in the contraction patterns of the SCM on the left and right sides of a healthy young person, making an otherwise symmetrical pattern asymmetrical [10]. B Valentino and coworkers have found a functional association between the masseter muscles and some leg muscles after occlusal dysfunction. In the test where cotton was placed between unilateral molars, the activity of the anterior tibialis remained constant, while the activity of the homolateral peroneus longus and the contralateral gastrocnemius increased [11]. In addition, Maurizio Bergamini et al. found that balancing the occlusion with acrylic wafers had beneficial effects on the balance of the sternocleidomastoid, erector spinae, and soleus muscles [12]. It can be speculated that asymmetrical occlusion may affect muscle contraction in different regions, and even postural modulation.

Currently, surface electromyography (sEMG) is commonly used for masseter muscle testing [13,14,15,16,17]. It has also become the widely accepted gold standard for checking the tone and symmetry of these muscles due to its noninvasive nature and ease of use [12]. This technique is now commonly used to verify the adjustment of dental occlusion and to check whether it affects the alteration of the resting state and the balance of important postural muscles.

Asymmetrical occlusion is common in patients with malocclusion, missing teeth, occlusal interference, chewing side preference, and so on. It has been proposed that the unilateral posterior crossbite affects static posture [18]. There is a significant difference in body postural stability between patients with myogenic temporomandibular (TMD) disorders and healthy controls [19]. The current studies on the effect of asymmetrical occlusion on body posture mostly concern healthy people or patients with orthodontic problems or TMD, but there are few studies on the effect on CSP, which has a high incidence in the population, and the survey found that its prevalence was 69%, 83%, and 76% during periods of deciduous, replacement, and permanent dentition, respectively [20]. Therefore, this study proposes comparing CSP subjects with control subjects to further analyze the specific effects of different occlusal conditions on muscle contraction in different parts of the body. Although some authors consider the lateral chewing habit as the side that first comes into contact with food, we prefer to define it as the side where most chewing cycles occur [21]. The long-term chewing unilateral preference is similar to the long-term repetition of the temporary asymmetric occlusion caused by the placement of cotton rolls on the unilateral posterior tooth; therefore, we designed this experiment. In the present study, we hypothesize that there is a difference in muscle symmetry between CSP and control subjects under different occlusal conditions. In this paper, MM, SCM, MGA, and LGA were selected as superficial muscle representatives to facilitate surface electromyography measurements, and muscle electromyographic activity was recorded under three different occlusal conditions: (1) bilateral posterior teeth biting cotton rolls; (2) left posterior teeth biting cotton rolls; and (3) right posterior teeth biting cotton rolls, to reveal the relationship between occlusion and postural muscle groups.

## 2. Materials and Methods

### 2.1. Participants

This study included 18 control subjects and 14 CSP subjects, who were students or staff of Shanghai Jiao Tong University School of Medicine and Shanghai Sixth People’s Hospital. All subjects gave informed consent for inclusion before participating in the study. The study was conducted in accordance with the Declaration of Helsinki, and the protocol was approved by the Ethics Committee of Shanghai Sixth People’s Hospital Affiliated with Shanghai Jiao Tong University School of Medicine (2023-KY-018(K)).

The inclusion criteria of the study were as follows: (1) age between 18 and 35 years old; (2) facial symmetry and complete dentition and no dental pathologies or dysfunction of the occlusion (including class I malocclusion); and (3) lateral chewing habit was included in the CSP group through questionnaires and a chewing gum test [22]; otherwise, these patients were included in the control group. The exclusion criteria were as follows: (1) class II or III malocclusion, history of orthodontic treatment, bruxism, or TMD; (2) oral and maxillofacial deformities or mandibular deviation; (3) presence of muscle pain or treatment with skeletal muscle relaxants or physiotherapy or joint injections within two weeks; (4) tumors or acute infections in the oral and maxillofacial regions or the spine and skeletal muscles; (5) abnormalities of the vestibular, proprioceptive, visual and nervous system; and (6) fractures, joint injuries, dislocations or other conditions that may affect body balance or the presence of significant postural problems [22,23].

### 2.2. sEMG Measurement

The sEMG activity of both sides of the MM, SCM, LGA, and MGA was recorded using an 8-channel surface electromyography instrument (MyoMove-EOW171039; Shanghai, China). The instrument was connected wirelessly to a computer, which displayed sEMG real-time changes as images and saved them automatically. Electrode pads (JK-1(A-H), Shanghai, China) were attached by the same rehabilitation physician with 5 years of clinical experience.

First, finger pressure was applied to observe muscle direction, and the skin surface was wiped with 75% alcohol cotton to reduce impedance. After natural drying, disposable self-adhesive electrode pads were placed on the skin directly above the muscle belly parallel to the direction of the muscle fiber, the distance between the centers of the two electrode pads was 2 cm, and the skin on the surface of the radial tuberosity of the right hand was chosen as the reference electrode (Figure 1) [24,25].
Masseter muscle: the subject was required to clench their teeth with force when cotton rolls were placed unilaterally or bilaterally in the posterior teeth;Sternocleidomastoid: the subject was asked to flex the neck with force;Lateral and medial gastrocnemius: the subject was required to plantar flex with force and maintain body balance.

**Figure 1 healthcare-11-01718-f001:**
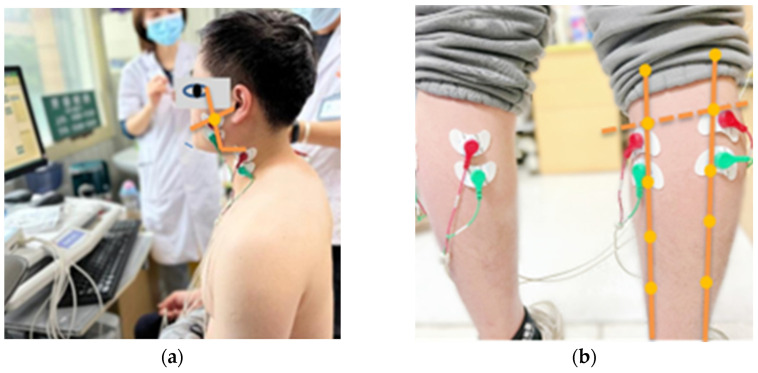
The upper edge of the electrode pads was placed: (**a**) masseter muscle, at the intersections between the labial-tragus lines and the gonion-exocanthion lines; sternocleidomastoid, at the same vertical height of the gonions; (**b**) lateral and medial gastrocnemius, at 1/5 of the distance of the shank from the knee joint line to the lateral and medial malleolus.

Subjects were asked to maintain Romberg posture (standing still with arms naturally positioned at the sides of the trunk) and to gaze at a 1 cm diameter circular mark on the wall at a distance of 90 cm with both eyes (Figure 2). The experiment was conducted in a quiet and undisturbed environment throughout [26]. The subjects were instructed to complete the above three movements simultaneously, the 5-s sEMG at the time of stabilization of each movement was recorded, and the middle 3 s were selected for statistical analysis. The amplitude of EMG activity (in μV) of each muscle was recorded in root mean square (RMS) and repeated three times for each occlusal position, with a 5 min break for each test. The EMG amplitudes of the MM, SCM, LGA, and MGA were measured in three different occlusal positions, as follows: bilateral (BCR), left (LCR), and right posterior teeth placement of cotton rolls (RCR) for clench with force. (The cotton roll is 19 mm long and 10 mm in diameter) [10].

### 2.3. Outcome Measures

The percentage overlap coefficient (POC%) was calculated to compare the symmetry of the left and right side muscles of each group in the CSP group and the control group [27]. $$\mathrm{POC}=[1-\frac{\sum \mathrm{Right}\mathrm{muscle}\mathrm{RMS}-\mathrm{Left}\mathrm{muscle}\mathrm{RMS}}{\sum \mathrm{Right}\mathrm{muscle}\mathrm{RMS}+\mathrm{Left}\mathrm{muscle}\mathrm{RMS}}]\times 100$$

The percentage difference in POC between unilateral and bilateral clenching cotton rolls [12] was calculated as follows:$$\%\;\mathrm{difference}\;=\;\frac{\left|LCR\;POC\;or\;RCR\;POC-BCR\;POC\right|}{BCR\;POC}$$

### 2.4. Repeatability Testing

Five subjects were randomly selected from among the subjects, recalled at two-week intervals, and tested for reproducibility by the same testers at the same period on the same experimental site. The intragroup correlation coefficient (ICC) was used to evaluate the reproducibility of bilateral muscle symmetry when biting cotton rolls bilaterally [28]. The ICC was 0.97 (*p* < 0.001) for POCSCM, 0.86 (*p* = 0.001) for POCMM, 0.99 (*p* < 0.001) for POCLGA, and 0.98 (*p* < 0.001) for POCMGA. The ICC of all measured variables was greater than 0.75, indicating that the sEMG measurements were reliable [29].

### 2.5. Statistical Analysis

SPSS version 26 (IBMCorp. Software, Armonk, New York, NY, USA) was used for statistical analysis. The sample size calculation was performed using PASS ver. 11 and was based on an anticipated standard deviation of 16 and a group difference of 13, according to the pre-experimental results. With a power of 80% and a two-sided α of 0.05, 14 participants were required per group [24]. Therefore, 32 eligible subjects were included in this experiment as the sample size.

The Independent samples *t*-test was used to calculate the differences in height, weight, and age between the CSP and control subjects. The independent samples *t*-test was used to assess significant differences in POC values for each pair of muscles between the CSP group and the control group. The measurements were expressed as the mean ± standard deviation. The paired *t*-test was used to compare the effect of different occlusal conditions on the sEMG of each group of muscles. The Spearman correlation coefficient was used to explore whether there was a correlation between the masseter muscle and other muscles. The statistical significance level of the hypothesis test was set at 0.05.

## 3. Results

A total of 18 control subjects (male = 8, female = 10) and 14 CSP subjects (male = 6, female = 8) were included in this study, including 8 with chewing left-sided preference (male = 1, female = 7) and 6 with chewing right-sided preference (male = 5, female = 1). There were no statistical differences in age, height, weight, and BMI between the two groups (Table 1).

In the control group, the symmetry of all superficial body muscles (MM, SCM, LGA, MGA) was not significantly different between males and females in all three occlusal positions, whereas in the CSP group, the symmetry of MM between males and females was different at BCR and RCR, not different at LCR, and the symmetry of other muscles and occlusal positions were not statistically different between the genders in the CSP group (Table 2).

There was a difference in the symmetry of the MM and LGA and there was no difference in the symmetry of the SCM and MGA between the CSP group and the control group at the BCR (*p* < 0.05). In addition, there was no significant difference in the symmetry of the SCM and MGA in all three occlusal positions between the CSP group and the control group (*p* > 0.05) (Table 3).

In three different occlusal positions, the symmetry of MM was significantly different in the CSP group. The symmetry of MM in the control group was significantly different when BCR was compared with LCR and LCR was compared with RCR, and was not different when BCR was compared with RCR. In three different positions, the symmetry of the SCM in the control group was significantly different when LCR was compared with RCR and BCR was compared with RCR, and there was no difference when BCR was compared with LCR. The symmetry of the SCM in the CSP group was significantly different when BCR was compared with LCR and LCR was compared with RCR and was not different when BCR was compared with RCR. In three different occlusal positions, the symmetry of the LGA and MGA was not significantly different in the CSP group and control group (Figure 3).

The percentage difference in symmetry of the SCM showed a positive correlation with that of the MM when the RCR was compared with the BCR (r = 0.415, *p* = 0.018). There was no significant correlation between the percentage difference in symmetry of the LGA and MGA and the percentage difference in symmetry of the MM.

## 4. Discussion

The purpose of this study was to compare the symmetry of the bilateral MM, SCM, LGA, and MGA between the control and CSP groups in different occlusal positions by using sEMG analysis, thus providing a preliminary investigation of the relationship between occlusion and posture.

There was a significant difference in the symmetry of the bilateral MM in different occlusal positions. In a study of the relationship between occlusal contact area and masticatory muscle symmetry, Napat Nalamliang et al. [30] found that the occlusal contact area was negatively correlated with the symmetry of the anterior temporalis muscle and the combination of the anterior temporalis and masseter muscles, but was not correlated with the symmetry of the masseter muscles alone. This seemingly contradictory conclusion virtually excludes the possible interference of the difference in occlusal contact area with the change in symmetry of the masseter muscles due to the artificial placement of cotton rolls in this experiment. In unilateral clenching cotton rolls, the RMS of the masseter muscle on the affected side is enhanced, i.e., the strength of the masseter muscle on that side is enhanced. This is consistent with the conclusion previously reported by Bakke, M and Mollor, E [31] that unilateral premature contact during maximal occlusion will result in significant asymmetry of the bilateral ascending jaw muscles, increasing homolateral muscle activity while decreasing contralateral muscle activity.

This also explains the worse symmetry of the masseter muscles in the CSP group than in the control group at BCR. The long-term chewing unilateral preference is similar to the repetitive training of experimental simulated asymmetric occlusion, which corresponds to more exercise of the masseter muscles on the early contact side and, therefore, results in increased muscle strength on that side. In this experiment, since there was slightly more chewing left-sided preference than right-sided, it can be assumed that the overall sample represents chewing left-sided preference. In the three different occlusal positions, the symmetry of LCR was the worst, BCR was better, and RCR was the best. This suggests that for those with a chewing left-sided preference, the use of bilateral occlusion is not the most effective way to restore muscle symmetry and that the use of contralateral occlusion can better restore balance. However, this is limited to experiments, and for clinical correction of masseter symmetry problems, the degree of recovery cannot be judged in real time. Therefore, the use of symmetrical occlusion can avoid the deepening of asymmetry to some extent.

In this study, it was found that in the control group, the symmetry of the measured superficial body muscles (SCM, MM, LGA, MGA) did not have differences between genders in all three occlusal positions. However, in the CSP group, the symmetry of MM differed between males and females at BCR as well as RCR. In the CSP group, there was only one male and seven females with a chewing left-sided preference, and five males and one female with a chewing right-sided preference. Thus, the overall sample of males in the CSP group was right-sided and the overall sample of females in the CSP group was left-sided. Therefore, the difference in symmetry of the masseter muscles between males and females at BCR reflects the significant difference in the original muscle strength between males and females. The degree of muscle asymmetry was also greater in males than in females. Therefore, when males representing chewing right-sided preference and females representing chewing left-sided preference were compared, LCR resulted in an increase in muscle strength on the left side of both sexes, thus offsetting the original strength difference between males and females, while RCR highlighted this gender difference.

The symmetry of MM and the symmetry of LGA were worse in the CSP group than in the control group at the BCR. There was more chewing left-side preference in the CSP group. The RMS of the left muscle was higher than that of the right, indicating that the muscle strength of MM and LGA is stronger on the chewing preference side than on the opposite side. This is consistent with B Valentino and F Melito’s experimental conclusion of a functional relationship between masticatory and leg muscles [11]. However, their conclusion that, with sterile cotton placed in the unilateral molar, the EMG of the tibialis anterior muscle remained unchanged while that of the homolateral gastrocnemius longus and contralateral gastrocnemius increased, seems to be inconsistent with the results of the present experiment. This is mainly because their experiment did not distinguish in detail between MGA and LGA. The gastrocnemius and soleus muscles make up the calcaneus triceps, which is an important muscle to keep the body upright. Thus, the balance of the occlusion is related to the stability of the upright body posture.

There was also a significant difference in the symmetry of the bilateral sternocleidomastoid muscles in different occlusal positions. This is consistent with the findings of Chiarella Sforza et al. [9] in a study of 11 male astronauts on the association between occlusion, sternocleidomastoid muscle, and body sway amplitude: as the position of the upper and lower jaws becomes more symmetrical, the contraction of the sternocleidomastoid muscle will also become more symmetrical and the body sway will subsequently decrease. In addition, the symmetry change of SCM was correlated with the symmetry change of MM when RCR was compared with BCR. This result supports the conclusion of V. F. Ferrario regarding the effect of asymmetric occlusal interference on the electromyographic activity of the SCM [10].

Thus, when the experiment caused temporary asymmetric occlusion, the change in symmetry of MM correlated only with the change in symmetry of SCM. However, when the CSP group and the control group were compared, the symmetry of MM and the symmetry of LGA at the BCR were significantly different between the two groups. It can be hypothesized that SCM plays the first regulatory role in instantaneous asymmetric occlusal changes but that long-term asymmetric occlusion may have an effect on the more distal leg muscles. The SCM has a dual innervation, with the accessory cranial nerve or XI entering the posterior triangle and innervating the SCM and trapezius [32]. The vestibular region is closely associated with SCM motoneurons, which rapidly improve posture and flexion of the neck when an external stimulus is induced; in addition, the cervical trigeminal reflex is directly related to the occlusal capacity of the temporomandibular joint and the EMG of SCM [33]. The physiological structure of the SCM could explain the observation that the SCM is associated with the change in symmetry of the MM in the case of the artificially created instantaneous asymmetric occlusion in this experiment.

The relationship between the stomatognathic system and posture can be explained by musculoskeletal and neuroanatomical influences [34]. Myers and Stecco et al. proposed that myofascial chains connect the body muscles as a whole and that tension in the contracted area has an effect on other areas near and far [35]. Tecco et al. found that an anterior cruciate ligament injury can produce changes in the MM, SCM, upper and lower trapezius, and anterior temporalis [36]. There are many anatomical connections between proprioceptive inputs to the stomatological system and neural structures (cerebellum, vestibule, etc.) related to posture [37]. P. Gangloff et al. showed that anesthesia of the command area of the unilateral trigeminal nerve (a type of sensory information for balance control) leads to a significant decrease in postural control in this area [38]. It is hypothesized that altered occlusion leads to a disruption of the myoelectric system of receptors in the masticatory muscles, periodontal ligament, or TMJ, which are uploaded through the trigeminal nerve and interconnected with multiple nuclei in the brainstem so that complex neural reflex activation leads to activation of long muscle chains, which interfere with postural control.

Occlusal interference, whether natural or experimental, can disrupt the function of the mandibular muscles and TMJ. In particular, experimental interferences, which often simulate and amplify dental occlusal characteristics, almost always have an immediate impact and then adapt relatively quickly to the changed conditions. Asymmetrical malocclusions, such as unilateral posterior crossbite, can also lead to altered neuromuscular coordination and abnormal myoelectric patterns in the masticatory muscles [10]. The association between occlusion and posture is confirmed to some extent by the present study, but the exact mechanism needs to be proven by further studies. There were some limitations in our study. First, since it was a preliminary study, the sample size of this study is small and there may be a high risk of interpretation bias, and it is necessary to be careful when interpreting the research results, and further in-depth studies with expanded sample sizes are needed to validate these results. Moreover, the subjects included in this study were all healthy adults, and the inclusion of patients who were forced to chew unilaterally for pathological reasons and subjects of different ages will be considered in the future to enrich the generalizability of the data, especially considering the inclusion of growing patients, as asymmetric mastication can influence the growth and development of children. Finally, the research method in this study is relatively single, and a combination of methods should be used to assess the relationship between occlusion and posture in the future, aiming to solve the whole-body posture problem through the stomatological perspective. In conclusion, CSP not only leads to dental caries, TMD, malocclusions, and asymmetrical facial development but also may have adverse effects on body posture. Therefore, for the CSP group, especially adolescents in the rapid growth stage, this unhealthy habit should be corrected as soon as possible to avoid negative effects on the whole body. In addition, this study also suggests that we need to pay attention to oral problems such as chewing habits and occlusion for patients with postural problems and provide better treatment plans for patients through a multidisciplinary approach.

## 5. Conclusions

The experiment-induced asymmetrical occlusion showed that the altered symmetry of the MM correlated with the altered symmetry of the SCM. Long-term asymmetrical occlusion (i.e., CSP) not only affects MM but also has potential effects on other superficial muscles (e.g., LGA).

## Figures and Tables

**Figure 2 healthcare-11-01718-f002:**
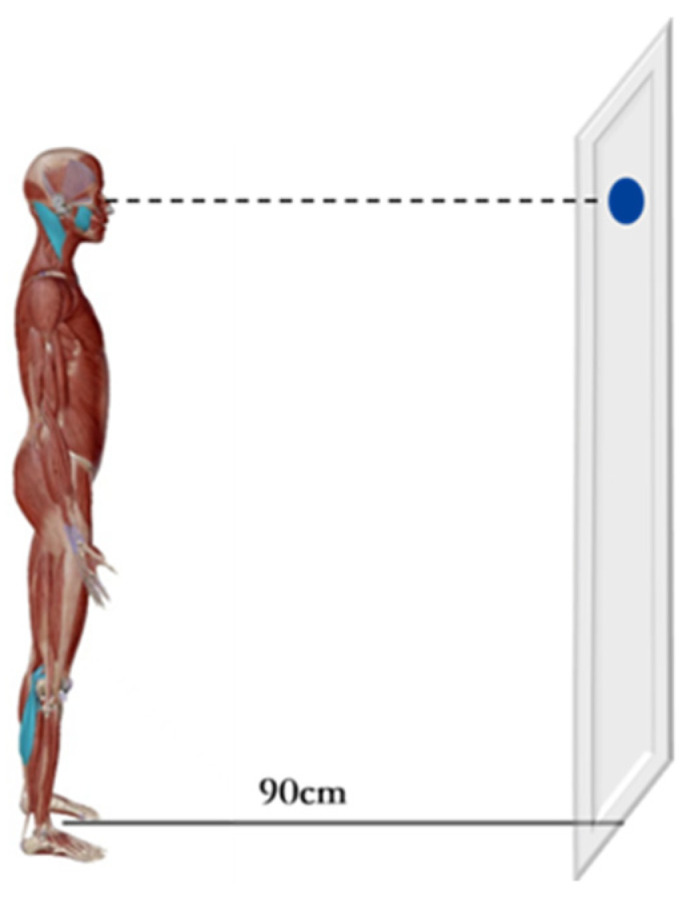
Schematic diagram of the experimental scene.

**Figure 3 healthcare-11-01718-f003:**
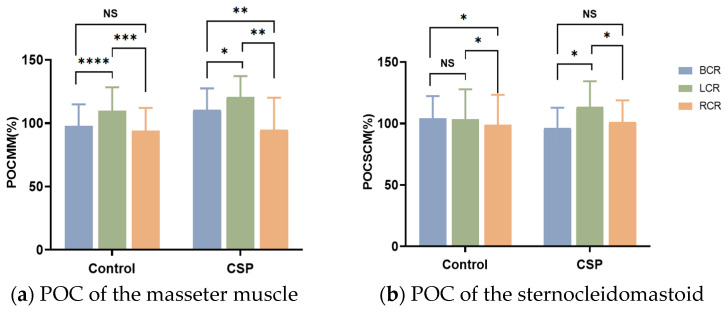
POC% of masseter muscles, sternocleidomastoid, lateral, and medial gastrocnemius under different occlusal positions. * Significant differences in POC between the control group and CSP group (*, indicating *p* < 0.05; **, indicating *p* < 0.01; ***, indicating *p* < 0.001; ****, indicating *p* < 0.0001). CSP for chewing side preference. BCR for bilateral, LCR for left, and RCR for right posterior teeth placement of cotton rolls for clenching with force.

**Table 1 healthcare-11-01718-t001:** Demographics of participants.

	Control Group	CSP Group
Number	Male = 8	Male = 6
	Female = 10	Female = 8
Age	26.6 ± 2.7	25.9 ± 2.4
Height (cm)	169.7 ± 9.8	167.9 ± 10.1
Weight (kg)	64.5 ± 11.2	60.1 ± 13.8
BMI (kg/m^2^)	22.2 ± 1.9	21.1 ± 2.7

Values are expressed as the mean ± standard deviation (SD). CSP for chewing side preference.

**Table 2 healthcare-11-01718-t002:** Percentage overlap coefficients (POC%) of the tested muscle pairs in the control group and the CSP group for each sex in different occlusal positions.

**Control**		**Male (POC%)**	**Female (POC%)**	***p* Value**
BCR	SCM	114.13 ± 27.41	97.44 ± 5.90	0.15
	MM	103.49 ± 19.61	93.50 ± 13.86	0.22
	LGA	103.21 ± 11.49	97.83 ± 14.49	0.40
	MGA	89.79 ± 25.7	99.84 ± 11.18	0.28
LCR	SCM	103.35 ± 32.61	103.21 ± 17.46	0.99
	MM	108.58 ± 33.51	106.81 ± 12.34	0.83
	LGA	98.36 ± 20.16	99.86 ± 5.40	0.87
	MGA	93.03 ± 27.41	100.02 ± 12.59	0.48
RCR	SCM	101.61 ± 34.05	96.36 ± 15.08	0.97
	MM	101.47 ± 15.98	88.14 ± 5.68	0.12
	LGA	96.17 ± 17.29	102.32 ± 15.54	0.44
	MGA	98.35 ± 17.43	93.99 ± 11.13	0.53
**CSP**		**Male**	**Female**	***p* Value**
BCR	SCM	87.58 ± 15.20	107.52 ± 22.91	0.09
	MM	96.32 ± 3.43	120.96 ± 4.82	0.00 *
	LGA	113.16 ± 25.71	111.65 ± 15.82	0.89
	MGA	98.19 ± 7.76	89.26 ± 17.57	0.27
LCR	SCM	105.19 ± 21.61	119.68 ± 6.81	0.23
	MM	114.73 ± 10.54	125.09 ± 19.27	0.26
	LGA	110.29 ± 25.57	107.71 ± 18.82	0.83
	MGA	103.38 ± 8.49	88.53 ± 14.37	0.05
RCR	SCM	85.32 ± 23.91	111.50 ± 20.96	0.05
	MM	77.94 ± 21.75	107.29 ± 20.77	0.03 *
	LGA	108.02 ± 27.97	109.79 ± 21.29	0.90
	MGA	103.21 ± 9.033	91.03 ± 14.84	0.10

POC values for each group are expressed as the mean ± standard deviation (SD). *, significant difference (*p* < 0.05) in the POCMM values between males and females at the BCR and RCR positions in the CSP group. SCM for sternocleidomastoid muscle. MM for masseter muscle. LGA for lateral gastrocnemius. MGA for medial gastrocnemius. CSP for chewing side preference. BCR for bilateral, LCR for left, and RCR for right posterior teeth placement of cotton rolls for clenching with force.

**Table 3 healthcare-11-01718-t003:** Percentage overlap coefficient (POC%) of the tested muscle pairs in the control group and CSP group in different occlusal positions.

	BCR (POC%)			LCR (POC%)			RCR (POC%)		
	Control	CSP	*p* Value	Control	CSP	*p* Value	Control	CSP	*p* Value
SCM	104.30 ± 17.90	96.11 ± 16.65	0.196	103.27 ± 24.48	113.47 ± 20.86	0.223	98.70 ± 24.60	100.99 ± 17.80	0.771
MM	97.94 ± 16.91	110.40 ± 16.97	0.048 *	109.79 ± 4.36	120.65 ± 16.46	0.095	94.06 ± 17.96	94.71 ± 25.32	0.933
LGA	100.22 ± 13.14	112.30 ± 19.74	0.047 *	101.42 ± 13.35	108.82 ± 21.07	0.235	99.58 ± 16.15	109.03 ± 23.36	0.186
MGA	97.60 ± 13.57	93.08 ± 14.50	0.372	98.02 ± 18.02	94.89 ± 14.04	0.596	95.93 ± 13.99	96.25 ± 13.75	0.949

POC values for each group are expressed as the mean ± standard deviation (SD). *, significant difference (*p* < 0.05) in the values of the POCMM and POCLGA at the BCR position between the CSP group and the control group. SCM for sternocleidomastoid. MM for masseter muscle. LGA for lateral gastrocnemius. MGA for medial gastrocnemius. CSP for chewing side preference. BCR for bilateral, LCR for left, and RCR for right posterior teeth placement of cotton rolls for clenching with force.

## Data Availability

The data are currently unavailable due to privacy restrictions.

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
