# Peer review of "The Effect of Asymmetrical Occlusion on Surface Electromyographic Activity in Subjects with a Chewing Side Preference: A Preliminary Study"

_healthcare, 2023, doi:10.3390/healthcare11121718_

Round 1
Reviewer 1 Report
Good job, it was a pleasure to read this paper.
However, some suggestions I would give are:
In general, achronyms should be checked.
About M&M,
Please, insert the reference for your SAMPLE SIZE.
In line 98, about "exclusion criteria" please quote one or both of these articles
Ugolini A, Mapelli A, Segù M, Zago M, Codari M, Sforza C. Three-dimensional mandibular motion in skeletal Class III patients. Cranio. 2018 Mar;36(2):113-120. doi: 10.1080/08869634.2017.1299830. Epub 2017 Mar 17. PMID: 28303737. Ugolini A, Mapelli A, Segù M, Galante D, Sidequersky FV, Sforza C. Kinematic analysis of mandibular motion before and after orthognathic surgery for skeletal Class III malocclusion: A pilot study. Cranio. 2017 Mar;35(2):94-100. doi: 10.1080/08869634.2016.1154681. Epub 2016 Apr 6. PMID: 27077258. Finally, please write a separate chapter for conclusions and check references again.
Thank you!
Author Response
Dear reviewer:
Thank you for your valuable comments! We have carefully considered the comments and suggestions for our manuscript. Appropriate changes have been made, and we have responded to questions raised point by point as follows:
Please see the attachment.
Yours sincerely,
Yubing Zhang

Reviewer 2 Report
Dear authors,
I really appreciate the methodology and the content of the paper. For my part I think there is a need to some basic research as presented in your article. Therefore, I think that the choice of the population for this paper took place in this way. All are quite in same age, young, healthy and with no dental reconstruction. It would be more interesting, if a more diverse population could be included. Who knows what the results would be like? I think this article lays the foundation for other analyzes to come.
Author Response
Dear reviewer:
Thank you very much for the supportive comment. We really appreciate it! In the future, we will do some basic research to further verify the conclusions we have obtained.
Yours sincerely,
Yubing Zhang

Reviewer 3 Report
First of all, I would like to thank the authors for the effort they have put into researching this topic. However, I would like to recommend a few considerations:
- According to the sections you have sent us, there is no "conclusions" section.
- Could you explain how you have calculated the sample size?
- In the inclusion and exclusion criteria, there is no reference to the facial pattern of the participants, which is key to evaluate the muscular activity of the patient. There are many studies that show that the activity of bachifacial patients is different from that of dolichofacial patients. It should also refer to patients suffering from bruxism. Muscle activity should be measured before starting the study to rule out the participation of some patients.
- In Table 2, the results obtained by women are, in most of the items, higher than in women in the CSP group. Have you taken this into account?
- In Table 3, there are hardly any statistically significant results. How could you interpret this detail at the clinical level?
- The present study is similar to a pilot study, as the differences found are small, therefore by increasing the sample size the values could change. How could I justify this point?
- In the discussion section it says that the aim of the study is as follows:
"The purpose of this study was to investigate the association between occlusion and body posture."
- However, the title of the study does not mention the concept "body posture". Do you think it should be modified?
- The sample are adult patients, do you think it is possible to carry out the study in growing patients, as asymmetric mastication can influence the growth and development of children.
Author Response
Dear reviewer:
Thanks for your review of our manuscript. The critical comments have greatly helped us in improving the quality of our manuscript. We have carefully considered the comments and suggestions for our manuscript. Appropriate changes have been made, and we have responded to questions raised point by point as follows:
Please see the attachment.
Yours sincerely,
Yubing Zhang

Reviewer 4 Report
The research question is interesting; however, the study is limited because small sample size and presentation of the data. A significant limitation is the small sample size. A Power analysis should be performed. Consider presenting this work as a case report.
In Material and methods, inclusion/exclusion criteria must be better defined (see PDF). The table2 must be presented in another way (see PDF). Figure 3 must be changed. In general, the legends must contain all information required to understand the presented data. Try must help the reader make it easy to read and follow.
Consider using consistent the word “control group”. Sometimes it is called "healthy" of "normal" group.
In the discussion, please state de limitations of your work.
Best regards,

Author Response

(The authors gave the same response as above.)

Round 2
Reviewer 3 Report
Dear Authors,
The article has been improved and the doubts answered. It is therefore suitable for publication.
Best regards
Author Response
Dear reviewer:
Thank you very much for the supportive comment. We really appreciate it!
Thank you and best regards.
Sincerely,
Yubing Zhang
Reviewer 4 Report
The patient population is too small. It should be clearly presented as a series of cases. It can be misleading. The shortcomings of this research are not presented in the discussion. Not only a small sample size but also conclusions can not be drawn.
Author Response
Dear reviewer:
Thanks for your review of our manuscript. We sincerely thank your valuable feedback that we have used to improve the quailty of our manuscript.We tried our best to improve the manuscript and made some changes to the manuscript, and we have responded to questions raised point by point as follows:
Please see the attachment.
Yours sincerely,
Yubing Zhang
